# An Overview on the Development of mRNA-Based Vaccines and Their Formulation Strategies for Improved Antigen Expression In Vivo

**DOI:** 10.3390/vaccines9030244

**Published:** 2021-03-11

**Authors:** Md. Motiar Rahman, Nan Zhou, Jiandong Huang

**Affiliations:** 1Institute of Synthetic Biology, Shenzhen Institute of Advanced Technology (SIAT), Chinese Academy of Sciences (CAS), Shenzhen 518055, China; nan.zhou@siat.ac.cn (N.Z.); jdhuang@hku.hk (J.H.); 2Faculty of Medicine, School of Biomedical Sciences, The University of Hong Kong, Hong Kong 999077, China

**Keywords:** IVT mRNA, mRNA vaccine, electroporation, protamine, LNPs

## Abstract

The mRNA-based vaccine approach is a promising alternative to traditional vaccines due to its ability for prompt development, high potency, and potential for secure administration and low-cost production. Nonetheless, the application has still been limited by the instability as well as the ineffective delivery of mRNA in vivo. Current technological improvements have now mostly overcome these concerns, and manifold mRNA vaccine plans against various forms of malignancies and infectious ailments have reported inspiring outcomes in both humans and animal models. This article summarizes recent mRNA-based vaccine developments, advances of in vivo mRNA deliveries, reflects challenges and safety concerns, and future perspectives, in developing the mRNA vaccine platform for extensive therapeutic use.

## 1. Introduction

Vaccines protect against millions of microbes and save thousands of lives from diseases each year [1]. Due to the use of extensive vaccines, the smallpox-causing virus (variola) has been eliminated and the cases of measles, polio, and various other childhood ailments have significantly declined throughout the world [2]. Traditional vaccine technologies including subunit vaccines and, live attenuated (or weakened) and inactivated (or killed) pathogens, offer long-lasting protection against various lethal ailments [3]. Despite this achievement, there exist key obstacles for the development of successful vaccines against diverse infectious disease-causing pathogens, particularly those better efficient to avoid adaptive immunity [1,4]. Besides, for the utmost evolving virus vaccines, the major difficulty is not the efficiency of established technologies but the demand for quick and large-scale production. Moreover, traditional vaccine technologies might be invalid for non-infectious disorders like cancer. Therefore, there is an urgent need for the progression of more versatile and effective vaccine platforms.

Nucleic acid-based treatments have developed as promising substitutes for traditional vaccine approaches. In animals, the first data of the use of successful in vitro transcribed (IVT) mRNA was reported in 1990, while reporter mRNAs were administered into in vivo mice model, and subsequently, protein expression was identified [5]. In 1992, a later study revealed that the use of messenger RNA encoding vasopressin in the brain hypothalamus can induce biological effects in rats [6]. Nonetheless, these primary but significant results could not make considerable investments in evolving mRNA-based vaccines, mainly due to concerns related to higher innate immunogenicity, mRNA instability, as well as inefficient drug delivery in vivo. Alternatively, the field approached DNA- and/or protein-based therapeutics [7,8]. Nonetheless, since the discovery of mRNA, it has been known the matter of consistent basic as well as applied research for many ailments [9,10]. In the initial decades of mRNA discovery, the main attention was on comprehensive investigations of structural as well as functional characteristics of eukaryotic mRNA and its metabolism. This is in order to make approaches for mRNA-based recombinant technology readily accessible to a wider research arena. In the late 1990s, preclinical investigation of IVT mRNAs was introduced for various applications, such as protein replacement and vaccination tools for cancer as well as infectious diseases [6,11,12,13,14,15,16,17,18]. Therefore, accumulated information allowed the latest scientific and technological improvements to overcome various difficulties related to mRNA, including its short half-life as well as adverse immunogenicity.

The administration of the mRNA-based vaccine has some benefits over subunit, live attenuated and inactivated virus, and DNA vaccines. Firstly, safety: there is no possible concern of insertional mutagenesis or infection since mRNA is non-integrating and non-infectious. Moreover, mRNA is generally destroyed under normal cellular conditions, and its in vivo half-life could be controlled via the use of diverse modification systems as well as delivery methods [19,20,21,22]. Besides, the innate immunogenicity of the mRNA could be downregulated to enhance the safety profile [19,22,23]. Secondly, efficacy: several modifications provide mRNA not only high stability but also translatability [19,22,23]. Efficient in vivo delivery of mRNA can be obtained by formulating them into carrier molecules such as polymers, peptides, lipid nanoparticles, micelles, allowing faster uptake and enhancing protein expression in the cytosol [20,21]. mRNA is considered the minimal hereditary material; hence, anti-vector mediated immunity is prevented, and these vaccines could be used recurrently. Finally, production: mostly due to the high yielding capability of IVT mRNA during transcription reactions, these vaccines have the potential for fast, low-priced, as well as scalable manufacturing.

The mRNA vaccine is a rapidly developing field; many preclinical investigations have been reported over the past few years [24,25,26,27,28,29,30]. Multiple human clinical trials have been initiated IVT mRNA-based therapeutics as protein-replacement therapy in the field of oncology [31,32,33,34], cardiology [35,36], endocrinology [37], hematology [38,39], pulmonary medicine [38,40], or the treatment of other diseases [14,41]. To further advance this revolutionary approach, unresolved issues like targeted mRNA delivery and its intricated pharmacology require to be developed.

Here, we review up-to-date mRNA-based vaccine approaches, describe the latest outcomes, highlight challenges and safety profiles in human trials, as well as recent successes. In addition, we demonstrate an inclusive summary of mostly used, protamine-, lipid nanoparticles- and electroporation/nanoparticle-based, drug deliveries, and their recent findings in vaccine development. Additionally, finally, the authors provide perceptions on the future opportunity of mRNA therapeutics. The reports propose that mRNA therapeutics can overcome various challenges in the development of vaccines for cancer and infectious ailments.

## 2. Structure of IVT mRNA for Improved Translation

Eukaryotic mRNA is generally transcribed in the nucleus, transported to the cytoplasm via nuclear export, and leading to protein synthesis. On the other hand, IVT mRNA must transfect into the cytosolic space from the extracellular matrix. Several factors are important for the successful delivery of IVT mRNA into the cells; (a) to overcome highly efficient ribonuclease of extracellular matrices, (b) to cross passive diffusion hindering cell membrane of the negatively charged mRNA macromolecules. Eukaryotic cells can engulf unprocessed mRNA very rapidly, however; the uptake efficiency and cytosolic transfer of IVT mRNA are very low in major cell types, which can be developed by modification of IVT mRNA with various complexing agents. This formulation prevents mRNA from ribonuclease-mediated degradation and facilitates cellular mRNA uptake. Apart from these, various techniques like electroporation could be used for transferring ex vivo mRNA rapidly into cells [9].

Significant efforts have been made to modify structural components of the in vitro transcribed mRNA, specifically the coding region, the 5′ cap, the poly(A) tail, and untranslated regions (UTRs) (both at 5′- and 3′ regions), to sequentially increasing its intracellular stability as well as translational efficiency (Figure 1). In fact, these advancements eventually lead to the synthesis of substantial amounts of protein over a longer time frame; ranging from a few minutes to longer than 1 week [42,43,44]. In the cytoplasm, substituting rare codons with recurrently used identical codons that show copious cognate tRNA is a usual habit to boost mRNA translation [45], while this model has been questioned in their accuracy [46]. Latest studies demonstrate that codon-optimization may disturb protein folding as well as function, enhance immunogenicity, and decrease efficiency. The analyses also reported that this strategy might develop difficulty in codon usage and challenges the scientific platforms for codon-optimization. Subsequently, codon-optimization might be unable to specify an optimal approach for boosting protein expression and may reduce the safety as well as the efficiency of mRNA-based therapeutics [46]. Improvement of G:C content represents an alternative sequence optimization that has been revealed to enhance steady-state mRNA concentrations in vitro [47] as well as protein expression in vivo [22]. While protein expression could be improved positively by inducing nucleosides modification, this modification might impact on the mRNA secondary structure [48], the kinetics and efficiency of protein synthesis and concurrent protein folding [49,50], and the induction of epitopes existing in other reading frames [46]. All these factors might possibly affect the level of immune induction. The kind of opportunities existing for modulating mRNA pharmacology is still unknown, and a greater level of understanding regarding mRNA-binding factors along with its binding sites might open a new door for engineering mRNA construct with various pharmacokinetic properties.

## 3. Effect of Nucleoside Modification in Antigen Expression 

Natural RNAs consist of adenosine triphosphate (ATP), guanosine triphosphate (GTP), uridine triphosphate (UTP), and cytidine triphosphate (CTP). In the case of nucleotide modification, a particular nucleoside undergoes modification after the transcription process. Owing to various limitations, presently, the easiest way to synthesize chemically modified RNA involves the in vitro RNA synthesis, in which a single nucleotide out of four fundamental nucleotide triphosphates (NTPs) is substituted with an analogous modified NTP [51]. In these primary transcripts, one selected base is replaced by an altered nucleotide at each location.

The latest analyses have revealed that the incorporation of naturally occurring nucleosides during post-translational modification has proven beneficial effects for offering low immunogenicity of IVT RNA [52]. For example, IVT mRNA having pseudouridine modification exhibited increased RNA stability and protein synthesis [19,51]. However, while RNA can induce the immune system via stimulation of toll-like receptors (TLR3, TLR7, and TLR 8) [53,54,55], incorporation of modified nucleosides into mRNAs, such as pseudouridine (Ψ), 5-methylcytidine (m5C), N6-methyladenosine (m6A), 2-thiouridine (s2U), or 5-methyluridine (m5U) (Figure 2), reduced its activity by considerably decreasing cytokine concentrations and biomarker efficacy in dendritic cells (DCs) [51]. This method, therefore, inhibits TLRs-mediated recognition and induces immunological defenses against the IVT mRNA [19]. Moreover, in bids to enhance and increase protein expression time, purification of IVT mRNA by HPLC was useful since this method eliminates dsRNA impurities, which caused lower production of IFN-1 as well as proinflammatory cytokines [23]. The Hartmann group further demonstrated that nucleoside modifications, for example, Ψ and s2U, prevented 5′-triphosphate RNA-induced stimulation of retinoic acid-inducible protein I (RIG-I), an alternative RNA-responsive immune sensor [52]. Hence, if any of the nucleoside modified IVT mRNAs would have translatability and also prevent in vivo innate immune activation, such RNA might be considered as a new therapeutic intervention not only for protein replacement but also for vaccination platform. To this line, Kariko et al. [19] investigated modified mRNAs in terms of their translational efficacy and immune properties in vivo. They reported that naturally occurring pseudouridine incorporation into mRNA protect RNA-induced immune activation both in vivo and in vitro and also increases the translational ability of the IVT RNA. These properties and the simplicity of producing such engineered mRNA by transcription reaction make mRNA a potential tool for the expression of any type of protein in either in vitro or cellular environment [19].

Studies have shown that nucleoside-modified RNA molecules encoding E glycoprotein and prM protein of Zika virus were encapsulated in LNPs (lipid-nanoparticles) [56]. Intradermal injection of a single low-dose of these proteins induced potent and long-lasting virus-neutralizing antibodies titer and therefore, protect against Zika virus challenges only by 30 and 50 μg mRNAs in mice as well as nonhuman primates, respectively. Nucleoside modification studies have also been demonstrated in recent studies using various antigens such as OVA [57], Luc; HIV-1 Env; ZIKV prM-E; and influenza virus HA [58], EBOV GP [59], luciferase (Luc), or erythropoietin (EPO), or scrambled EPO coding region (scramble) [60], etc.

## 4. Formulations of mRNA for In Vivo Drug Delivery

The mRNA is generally synthesized in a cell-free condition by transcription reaction from a DNA template. Since the mRNA candidates need to cross membranous lipids, various nanoparticles (protamine, LNPs, lipid polymers hybrid-, gold- and polymeric nanoparticles), and cell-based delivery were optimized as tools to load and deliver RNA into the cytosol [61]. This review summarized the application of APCs (antigen-presenting cells) and nanoparticles (protamine, LNPs) in mRNA delivery with recent examples of clinical trials.

### 4.1. Protamine

Protamine is a positively charged natural protein that has an outstanding ability to bind with nucleic acid-like mRNA (Figure 3) and improved its uptake and transfection ability [62]. It has been shown that positively charged protamine could efficiently form a complex with messenger RNA through electrostatic interaction [63] and this complex may act as a danger signal and stimulates murine cells via MyD88-specific signaling including TLR7 and TLR8 [62]. Hoerr and colleagues revealed that this complex destroys within 2 hours following the incubation in the serum sample, which restricts their capabilities for endurance in the in vivo circulation [17]. They subsequently demonstrated that incompletely degraded protamine/mRNA complex can still show immunostimulatory action for more than 100 h [62]. It was also reported that protamine-complexed mRNA intensely stimulated a variety of white blood cells (WBCs) including B lymphocytes, granulocytes, and natural killer cells, and notably, elicited overall immune response compared to that of protamine-complexed DNA [17,62].

Weide et al. [28] reported the efficiency of protamine/mRNA complex by experimental analysis of intradermally-inoculated protamine/mRNAs complex that encodes for a variety of enzymes, such as Tyrosinase (an enzyme that helps to catalyze melanin formation), Melan-A (a melanoma antigen), gp100 (a protein included in the maturation of melanosome), Survivin (a protein which involves in the maintenance of apoptosis) of metastatic melanoma patients, MAGE-A1 (a melanoma antigen) and MAGE-A3 (a melanoma antigen) [28]. No adverse effects were observed higher than mild to moderate (grade II level) and overall, a complete clinical response, such as substantial result on the rate of immunosuppressive cells and rise of antigen-specific T cells was witnessed.

CureVac (Tübingen, Germany) recently studied the use of protamine/mRNA complex. In vivo analysis has shown that an mRNA vaccine (a two-component vaccine, RNActive) consists of both free and protamine/mRNA complex evolved an effective antigen expression and induced TLR7-mediated immune stimulation [64]. An efficient adaptive immune response with both humoral and T lymphocytes-mediated immunity was also obtained which exhibited not only therapeutic efficacy but also prophylactic activity against a tumor. In a separate study, CureVac has explored an analysis on castrate-resistant prostate cancer patients using protamine-complexed mRNA encoding for prostate-specific membrane antigen (PSMA), prostate-specific antigen (PSA), six transmembrane epithelial antigen of the prostate 1 (STEAP1), and prostate stem cell antigen (PSCA) [65]. After intradermal injection, 80% of the patients showed immune induction against the administered mRNA antigen and 60% against multiple antigens, which eventually interrelated with prolonged survival [65].

Using protamine-complexed mRNA, the Petsch group [66] reported that intradermal injection of mice with two component-mRNA encoding full-length hemagglutinin from H1N1 influenza A virus (A/Puerto Rico/8/1934) stimulated efficient seroconversion along with virus-neutralizing antibody titers in all immunized mice. Immune induction was long-durable and protected animals against influenza A virus challenge of the H1N1, H3N2, and H5N1 strains [66]. The efficiency of the two-component CureVac vaccine was further reported in pigs and ferrets [67].

In 2016, CureVac explored the protamine-mRNA complex to induce an immune response against the rabies virus. In that study, non-replicating rabies glycoprotein (RABV-G) encoding mRNA was optimized to mediate potent virus-neutralization in mice and domestic pigs. The analysis reported an exceptional induced anti-rabies immunity in both animals. More notably, protamine-complexed RABV-G mRNA inoculated mice were protected from lethal intracerebral rabies infection. Consequently, the cellular and humoral immune responses noticed in non-human primates by RABV-G mRNA against rabies infection were superior over licensed vaccines Rabipur (LIC) and HDC [68]. Some of the currently used protamine-complexed mRNA vaccines with their clinical efficacy have been reviewed in Table 1.

**Table 1 vaccines-09-00244-t001:** Protamine-mRNA formulated vaccines.

Target mRNA	Stage	Findings	Ref.
β-gal and GFP	HeLa-K cells injected into B6 (H2) and BALB/c mice	Successful CTL response, dependent on injection site	[17]
β-gal or CMV pp65	Murine BM-DC	Stimulated mouse BM- DC: induced IL-6 and IL-12 release and up-regulation of CD86	[44]
β-gal, EGFP, or CMV pp65	Human PBMC	Complexes induced release of strong IL-6 and TNF-α, stimulation of innate immunity and other APCs	[62]
Melan-A, Tyrosinase, gp100, MAGE-A1, MAGE-A3, and Survivin	Individuals with metastatic melanoma	Raised frequency of immunosuppressive and vaccine-directed cellular immune response	[28]
OVA (GgOVA), control vaccine (Ecβ-gal sh), PSMA (HsPSMA), and STEAP vaccine (HsSTEAP)	Rat (C57BL/6, BALB/c)	Showed antitumor by activating adaptive and innate immune systems, stimulation of toll-like receptor 7 (TLR-7), ability to inhibit established tumors, induction of two component mRNA vaccine	[64]
Ovalbumin with radiation, two component vaccine	Rat (C57BL/6)	mRNA immunotherapy and tumor irradiation act synergistically to eradicate established tumor (Lewis lung cancer)	[69]
Rabies glycoprotein (RABV-G)	Rat (C57BL/6, BALB/c) and domestic pigs	Induced potent neutralizing antibody superior to licensed vaccines, induced lethal challenge against rabies, induce homeostasis	[68]
RNActive Ovalbumin, luciferase fused rabies glycoprotein, two component vaccine	Rat (C57BL/6, BALB/c)	Vaccine taken up by leukocyte and non-leukocytic cells, represented by APCs, transport to draining lymph nodes (dLNs), T-cell proliferation, immune cell activations, and induction of adaptive immunity	[70]

### 4.2. Lipid Nanoparticles

The area of lipid nanoparticles-based vaccination is comparatively advanced than other methodologies. Until now, various lipid-based formulations have been investigated for their efficiency in mRNA delivery. Among these, cationic lipids are widely administered due to their promising electrostatic interactions with anionic mRNA to form nanobiomaterial. LNPs show many advantages over other vectors, such as (a) the synthesis of LNP is robust, where both constituents and composition can easily be changed to improve delivery efficacy with lower toxicity, (b) LNPs have been effectively used earlier as delivery vehicles for mRNA-based vaccines, and (c) immune potentiators, including adjuvants or immune cell directing ligands, can be included to modify the immune response [57,71,72,73]. This field was pioneered in 1989 by Malone et al. while DOTMA (N-[1-(2,3-dioleyloxy)-propyl]-N,N,N-trimethylammonium chloride) was used to transfect human, drosophila, rat, Xenopus (frog), and mouse cells with mRNA encoding luciferase [74]. However, the clinical progress of such lipid-based tools has been troubled by their toxicity [74]. Despite these limitations, DOTMA together with DOTAP (1,2-dioleoyloxy-3-trimethyl ammonium propane chloride) showed a promising candidate for this purpose [75]. Moreover, although cationic LNPs can be efficient in vitro; however, in vivo effects are not much satisfactory because positively charged liposomes can readily be removed by the mononuclear phagocytosis.

The overlay of poly-ethylene-glycol (PEG) on lipid carriers has been applied extensively as delivery vehicles for nucleic acid payloads, including DNA [76] and siRNA [77] to develop formulation method, decrease aggregation, and increase the time of blood circulation [78,79,80]. However, the PEGylation layer has also been exhibited to decline cellular uptake, an outcome that might be reduced by optimizing the content and size of PEG [76,81,82]. PEGylation has also been used for the delivery of nanoparticle-formulated mRNA, especially for lipid-polymer hybrid- and polymer nanoparticles [83,84]. The use of PEG modification to LNPs-based mRNA delivery systems for the development of their in vivo efficiency remains elusive. Incorporation of 1,2-dioleoyl-sn-glycero-3-phosphoethanolamine (DOPE), a well-known helper lipid, is another approach to decrease accumulation of the lipid systems and to develop endosomal release (Figure 3) [85,86,87]. Akaike et al. [88] developed an alternative approach to further improve mRNA transfection efficiency in both non-mitotic and mitotic cells, by coating inorganic carbonate apatite nanoparticles on liposomal transporters. These inorganic apatite nanoparticles were shown to improve mRNA uptake via efficient endocytosis. They further reported that coating mRNA-containing DOTAP-apatite nanoparticles with arginyl-glycyl-aspartic acid, which is identified for its capacity to complex with integrins [89], increased cytoplasmic expression of transported mRNA. Owing to its potency for targeting as well as effective delivery of mRNA, this method seems to be versatile and bears huge potential [75,88,90,91]. Very recently, a new lipid/protamine/mRNA nanoparticle system was planned and broadly applied for systemic delivery to tumors [31]. DOTAP liposomes have been explored in this system to encapsulate protamine-complexed mRNA and later coated with DSPE-PEG and DSPE-PEG-anisamide [31]. This platform showed good stability from degradation in serum, elevated in vitro transfection efficiency in NCI-H460 cells, very low cytotoxicity, deposition in in vivo tumor site as well as anticancer activity [31].

Until now, LNPs have been extensively applied, predominantly to pioneer mRNA to induce immune cells for vaccine purposes. CureVac formulated mRNA with several sizes of (70~100 nm) lipid nanoparticles prepared by ionizable amino lipid, phospholipid, cholesterol, and PEGylated lipid. The developed vaccine was shown to be well-tolerated in NHPs (non-human primates), including mice and pigs, and elicited long-lasting humoral immune responses, which correlated with the protection against influenza and rabies infection. Remarkably, the cellular and humoral immunity in NHPs, mediated by LNP/mRNA vaccines, against influenza H3N2 and rabies viruses were superior over licensed Rabipur and Fluad vaccines, respectively [92]. The copious research conducted by CureVac (Thress et al.) using three different Epo (erythropoietin) sequences engineered, but chemically unmodified nucleoside mRNA formulated in LNPs (ionizable cationic lipid/phosphatidylcholine/cholesterol/-PEG; 50:10:38.5:1.5 mol/mol) [22]. The Epo mRNA vaccine resulted in significant physiological responses in mice and NHPs. Even in about 20 kg weight pigs, a single sufficient dose of engineered mRNA formulated in LNPs produced high systemic Epo levels and solid physiological outcomes.

Bahl et al. [93] formulated two H7N9 HA and H10N8 HA mRNA vaccines of influenza A in lipid nanoparticles (ionizable lipid: 1,2-dis-tearoyl-sn-glycero-3-phosphocholine: PEG: cholesterol) at 50:10:1.5:38.5 molar ratios. In this study, it was shown that LNPs complexed-modified mRNA encoding HA of H10N8 (A/Jiangxi Donghu/346/2013) or H7N9 (A/Anhui/1/2013) elicited quick and strong immune responses in ferrets, mice, and NHPs, as demonstrated by microneutralization and hemagglutination inhibition assays. Notably, a single dose of mRNA encoding H7N9 HA protected mice against the lethal challenge and lessened lung viral titers in ferrets. Results from an escalating-dose of phase 1 H10N8 HA trial first-in-human, exhibited very high rates of seroconversion, indicating a robust prophylactic immune response. Mild or moderate adverse effects were observed with no serious or only a few severe events. These data demonstrated that LNP-complexed modified mRNA can introduce adaptive immunogenicity with a satisfactory tolerability profile.

In a separate study, a single dose of LNP-mRNA was demonstrated to induce rapid, robust, and long-time immunity in vivo; hence, facilitating both therapeutic and prophylactic challenge from lethal rabies intoxication or botulinum infection. Besides, therapeutic mRNA-induced antibody expression helped mice to survive against a lethal tumor [94]. A comparative analysis between pseudouridine-modified and unmodified mRNA-formulated C12-200 LNPs was performed in the in vivo mice model by Kauffman et al. [60]. Pseudouridine-modified mRNA showed no substantial benefit on physical properties of lipid nanoparticle, in the in vivo protein expression, or mRNA immunogenicity over unmodified mRNA while inoculated systemically with liver-directing LNPs, but decreased in vitro transfection efficiency. This report gives an insight into LNPs/modified-mRNA mediated immune responses and advocates that pseudouridine modifications might not be essential for LNPs/mRNA-based therapeutics in liver disease.

Anderson et al. [57] reported a library to develop optimal lipid nanoparticle composition by varying the formulation parameters (Table 1 of [57]). The efficiency of the clinical trial was examined in an aggressive model of B16F10 melanoma. The analysis demonstrated a robust CD8^+^ T cell stimulation after a prime vaccination induced by the optimal composition of LNPs/mRNA. Treatment of B16F10 melanoma with LNPs complexed mRNA encoding for TRP2 and *TAAs* gp100 resulted in tumor fall and prolonged the overall endurance of the treated mice. The report also demonstrated increased immune response by the incorporation of lipopolysaccharide, a well-known adjuvant.

The Anderson group further established a combinatorial library using ionizable lipid-like materials to act as mRNA delivery vehicles, which accelerate the in vivo mRNA delivery and induce strong and explicit immune activation. Using a 3D multi-component model system, over 1000 lipid structures were synthesized and evaluated for their potency. The top formulations lipid candidates elicited a strong immune response and were shown to reduce tumor progression and increase survival in human papillomavirus E7 and melanoma. These lipid candidates share a corporate structure, such as, an unsaturated lipid tail, a dihydroimidazole linker, and cyclic amine head groups (Figure 3). Such formulations of lipid nanoparticles stimulated APCs maturation through the intracellular stimulator of interferon genes (STING) pathway, rather than via TLRs, and resulted in lower expression of systemic cytokine and improved anti-tumor efficacy [95].

A comprehensive overview of existing LNPs-based mRNA deliveries has been presented in various contemporary analyses [96,97,98,99,100,101], and a summary of these novel innovations has been shown in Table 2. Since the LNPs demonstrate a more comprehensively used system, their further improvement and optimization might open up a new door for the invention of more efficient mRNA delivery platforms.

**Table 2 vaccines-09-00244-t002:** LNPs-mRNA formulated vaccination.

Target mRNA	Lipid Nanoparticle Contents	Stage	Findings	Ref.
Luciferase	DOTAP liposomes covered with apatite nanoparticles	HeLa	Along with ARCA had more than 100-fold increase compared to DOTAP, proportion not assessed	[91]
Luciferase	DOTAP liposomes protected with apatite nanoparticles	HeLa NIH 3T3	9–14 fold improved compared to mRNA liposome alone, proportion not determined	[90]
Luciferase	Fibronectin associated DOTAP liposomes protected with apatite nanoparticles	HeLa	Fn-DOTAP-apatite complex showed 50-fold increase than DOTAP alone, proportion not assessed	[75]
TriMix mRNA encoding CD40-ligand, CD70 and TLR	DOTAP/DOPE/DSPE-PEG-2000-biotin	Primary murine bone marrow-derived DC from C57BL/6 mice	19% improved	[102]
Luciferase	DOTAP/DOPE/DSPE-PEG-2000-biotin lipoplex loaded microbubbles	DC primary cultures from the bone marrow of C57BL/6 mice	24% improved	[103]
EGFP	Lipofectamine 2000 and TransIT	Neurospheres from subventricular zone of adult C57BL/6 mice	40–50% improved	[104]
GFP and luciferase	MLRI/DOPE and TransFast	CHO, NIH3T3	>50% improved >40% improved	[105]
EGFP, B-16	Novel cationic lipids: X2, S1, S2, S3, 2X3, and 2D3 with DOPE	DC cells cultured from the bone marrow of C57BL/6 mice	Up to 47% of DC progenitors Up to 57% of immature DCs	[106]
Herpes simplex virus 1-thymidine kinase	DOTAP-cholesterol liposome with DSPE-PEG and DSPE-PEG-AA, encapsulating protamine-mRNA cores	NCI-H460 xenograft	68~78% improved	[31]
GFP, Luciferase and CXCR4	DOTAP/DOPE	HeLa	~80% improved	[107]
Luciferase and GFP	Stemfect	JAWS II DC2.4	80%; >97%; >50% and >60%	[108]
*Photinus pyralis* luciferase (PpLuc), rabies glycoprotein (RABV-G), influenza	(70~100 nm) lipid nanoparticles prepared by ionizable amino lipid, PEGylated lipid, phospholipid, and cholesterol	BALB/c, pigs	Lipid formulated mRNA vaccine induced protective antibody titers; boosted and stable for 1 year	[92]
*Photinus pyralis* luciferase (PpLuc), Epo (mouse, pig and maque)	Inonizable cataionic lipid/phosphatidylcholine/cholesterol/PEG; 50:10:38.5:1.5 mol/mol	HeLa, BALB/c, pigs, monkeys	Induced high mRNA expression and elicited significant physiological response in mice and nonhuman primates	[22]
mRNA encoding hemagglutinin of H10N8 (A/Jiangxi-Donghu-/346/2013) or H7N9 (A/Anhui/1/2013) influenza virus	Ionizable lipid: 1,2-distearoyl-sn-glycero-3-phosphocholine (DSPC): cholesterol: PEG-lipid (50:10:38.5:1.5)	HeLa, BALB/c, ferrets, cynomolgus monkeys, human	Induced rapid and robust immune responses in ferrets, mice, and NHPs; single dose of mRNA encoding H7N9 saved mice against lethal challenge and decreased lung viral titers in ferrets; elicited robust immune response in humans with mild or moderate adversity	[93]
Luciferase, Ovalbumin (OVA) expressing B16F10 mouse melanoma	Lipid nanoparticles library	C57BL/6J	Optimized LNPs showed transfection in various immune cells; stimulation of a robust CD8^+^ T-cell response after single immunization; greater survival rate in a transgenic mice melanoma	[57]
Firefly luciferase, Ovalbumin (OVA) expressing B16F10 mouse melanoma, papilloma E7 protein	Multi-dimensional over 1000 lipid nanoparticles consisting of heterocyclic ring	HeLa, bone marrow-derived dendritic cells and bone marrow-derived and peritoneal macrophages, Ai14 mice model	Top-performing lipid elicited a robust immune activation, prevented tumor progression and long-lasting survival in human papillomavirus E7 and melanoma in the in vivo tumor model	[95]

### 4.3. Electroporation Plus Nanoparticles Formulation

Cell-specific mRNA delivery (electroporation) represents an alternative approach for the expansion of mRNA vaccines. This can increase mRNA delivery to the target cells and thus decrease necessary mRNA dose by decreasing possible off-target effects. The cell-specific mRNA delivery works on the principle of professional APCs (e.g., dendritic cells) being in the vicinity of T-lymphocytes in these cell organs, therefore, offering optimal environments for effective priming as well as increasing T-cell stimulations in vivo (Figure 4) [71,109]. However, previous observations have shown that DCs have been only and weakly transfected by lipoplexes [100]. Hence, nanoparticle formulations were needed to optimize for improved targeting of dendritic cells [84].

The cell-specific delivery of IVT mRNA/nanoparticles through effective targeting has been reported to induce potent effector as well as memory T-lymphocytes responses, and strong IFN-α-mediated eradication of advanced tumors. Associated with cancer immunotherapy, dendritic cells could be transfected with either total tumor RNA or tumor-associated antigens (TAAs) encoding mRNA [109]. DCs transfected with messenger RNA encoding tumor-associated antigens can be used directly for vaccination purposes without the demand of exploiting patient-specific tumor-derived cells or protein antigens [13,110,111]. The shortcomings of this method include the deficiency of known TAAs for various cancers and the choice of TAAs might be challenging since not all recognized TAAs induce antitumor immunity. Experimental analyses on TAA mRNAs were shown to prompt the induction of antitumor immunity [112]. For example, dendritic cells transfected with mRNA encoding prostate-specific antigen (PSA) TAAs induced potent PSA-specific T-cell responses in prostate cancer patients and elicited a substantial decline in PSA level in 6 out of 7 patients [24]. Furthermore, inoculation with CEA (carcinoembryonic antigen) mRNA transfected DCs revealed a well-tolerance in pancreatic cancer individuals while antitumor immunities were only achieved in 6 of 24 patients [25,26]. Besides, mannosylated-histidylated-lipopolyplex were loaded for improved mRNA transfection of dendritic cells [84]. Intravenous injection exhibited four-times lower DCs expressing EGFP for mRNA-loaded with sugar-free LPR100 compared with Man (11)-LPR100. The increased transfection of DCs associated with increased growth inhibition of B16F10 melanoma and prolonged survival period after vaccination with MART1 mRNA-formulated Man (11)-LPR100. The method of exploiting total tumor RNA from tumor-specific patients was also assessed in clinical trials for renal [113,114], lung [115], brain [116] cancers, and melanoma [117,118,119]. In this regard, clinical findings to neuroblastomas and brain tumors were witnessed in approximately a third of the total registered patients [116]. Additionally, analyses in individuals with renal cell carcinoma exhibited no indication of induced autoimmunity or dose-limiting toxicity [113]

## 5. Challenges and Safety Issues in the Development of mRNA-Based Vaccines against Novel Antigens

The potential benefit of RNA-based therapeutics is their rapid development with lower side effects. The single-stranded IVT mRNA therapeutics are free from the hazard of genomic integration with the host cell and are efficient to generate high-quality viral protein. Besides, mRNAs are rapidly expressed and thus allowing the protein to be manufactured inside the cell. Experimental analyses of mRNA for rapid biological therapy has revealed outstanding tolerability as well as safety profile and reported that mRNA-based vaccines have no potential platform-inherent concerns [120]. However, for a majority of other mRNA-based therapeutic applications, such as protein replacement therapy, where preclinical and clinical trials are still very limited, researchers are unclear about the potential challenges and types of safety issues that should be considered [121]. Nevertheless, vaccine developers have pointed out several challenges and safety trials for developing novel vaccines against new infections e.g., SARS-CoV2.

Like other viruses, the novel coronavirus is an RNA virus, which exerts its effect via ‘S’ protein consisting of 1273 amino acid residues. This virus showed high mutation efficacy and genetic instability that may hamper immune induction [122]. Therefore, it is vital to understand genetic changes in the coding as well as non-coding sequences, genetic variant, pathogenicity, and host–pathogen interrelations. Genetic alteration in the ‘S’ protein likely to stimulate its folding pattern, which may change antigenicity and, consequently, may disturb vaccine design [123,124,125]. Copious reports advocated that mutations in the target proteins may be interrelated with drug resistance, leading to vaccine inefficacy. Therefore, immunogen selection for the mRNA-based vaccine development should be carefully considered and designed.

It is also critical to focus that insecurity over long-durable protection still exists against COVID-19. In several reports, there is evidence of reinfections. In such cases, it is required to report how long a protective immune induction will be continued in a patient [126,127,128]. Regarding the immunogenicity of the COVID-19 mRNA therapeutic in elderly individuals, it has been demonstrated that after the second inoculation, serum neutralizing titer was noticed in all the populations. Surprisingly, the binding as well as antibody neutralizing efficiency was comparable to those stated among vaccine receivers from 18 to 55 years old and were over the average of a group of control populations [129].

In the present pandemic situation, several researchers advised that immune inductions against coronavirus can lead to ADE (antibody-dependent enhancement) [130,131]. Although it was reported that immunization of the COVID19 receptor-binding domain did not mediate ADE in rodents [132], this principle could contribute to the pathology of several feline coronaviruses and flavivirus [132], notably dengue virus [133]. Nevertheless, ADE should be taken into consideration while evolving therapeutics against novel viruses [134,135]. In addition, considering the emergency need for a COVID-19 vaccine worldwide, being over-precautions, the authority should not stop the release of safe, well-tolerated, and efficient vaccines to the general populations [135,136,137]. Some reports demonstrated the safety concern of vaccine-enhanced disorder for inactivated vaccine candidates, remarkably vaccine-associated enhanced respiratory disease (Table 3) [138,139,140,141].

Although mRNA vaccines can be manufactured with a minimum time; however, large-scale production of these therapeutics remains a challenging task owing to its huge uncertainty to meet the demand during the pandemic. Additionally, in recent years, nucleic acid-based vaccines could not produce effective platforms for human infections and other diseases using temperature-sensitive lipid nanoparticles, which may hamper for scaling up vaccine manufacture [138]. For example, BNT162b2 and the Pfizer-BioNTech COVID-19 vaccines, are lipid nanoparticle-formulated nucleoside-modified mRNA vaccines that encode SARS-CoV-2 spike protein [121]. Following the issuance of the emergency use authorization (EUA) for the BNT162b2 vaccine by the U.S. Food and Drug Administration in December 2020, the vaccination scheme of this vaccine was launched later in the U.S and other countries. However, the plan for massive immunization has been delayed by the stringent requirement of storage and transportation of the BNT162b2 vaccine. Besides, cold chain transportation is not available in many COVID-19 epidemic areas, so potent mRNA vaccines with enhanced temperature stability will be highly preferred in the future. Despite huge challenges and risk factors involved in the development of mRNA-based vaccines in clinical trials, several mRNA-based vaccines in human trials have been reported in the literature that has been summarized in Table 3.

**Table 3 vaccines-09-00244-t003:** mRNA-based vaccines in human clinical trials with their major finding and adversity.

Antigen/Study Identifier/Phase	Subjects/Numbers	Route	Major Findings	Ref.
Rabies glycoprotein/NCT02241135/Phase I	18–40 years (volunteers), 101 healthy individuals	ID and IM	94% of ID and 97% of IM vaccinated populations received severe injection site reactions, and 78% ID and 78% of IM injected peoples demonstrated severe systemic reactions, induce antibody response when administered with a needle free device, safe with a tolerability profile	[142]
Melan-A, Tyrosinase, gp100, MAGE-A1, MAGE-A3, Survivin/NCT00204607/Phase I/II	18–80 years, 21 patients with metastatic melanoma	ID	No adversity was observed more than grade II, feasible and safe, rate of Foxp3+/CD4+ regulatory T lymphocytes were reduced significantly upon mRNA plus keyhole limpet hemocyanin (KLH) injection, CD11b+HLA-DR lo monocytes (myeloid suppressor cells) were decreased in the patients without KLH addition	[28]
NY-ESO-1, MAGEC1, MAGEC2, 5T4, Survivin, MUC1/NCT01915524/ Phase 1b	≥18 years, 19 patients with NSCLC	ID	No serious toxicity was observed, only 7% patients experienced grade >3 related adversity, antigen-mediated immune induction was seen in more than 2/3 of patients	[143]
HIV-1/NCT00672191/Phase II	18 to 60 years, 59 participants	ID	Develop immune control of HIV-1 reproduction	[144]
Spike protein (COVID-19)/NCT04470427/Phase II	18 to 99 years, 30,000 participants	IM	Ongoing	[145]
Spike protein/NCT04283461/Phase I	56 to 70 years, 40 healthy adults	IM	Mild or moderate adversity was observed, 100 μg mRNA produced higher virus neutralizing-antibody titers than 25 μg	[129]
Spike protein/NCT04368728/Phase I and II	18 to 55 years, 45 adults	IM	Adversity was dose-dependent, transient, mostly mild to moderate	[146]
Spike protein/NCT04283461/Phase I	18 to 55 years, 45 healthy adults	IM	This vaccine candidate induced immune responses against COVID-19 in all populations, and no trial-limiting safety issues were detected	[147]
Spike protein/NCT04566276/Phase I and II	65 to 75 years, 600 healthy adults	IM	Ongoing	[148]
Spike protein/NCT04515147/Phase II	18 to 60 years, 691 participants	IM	Ongoing	

ID = intradermal; IM = intramuscular; NSCLC = non-small cell lung cancer.

## 6. Future Direction and Conclusions

mRNA-based therapies are state-of-the-art and rapidly evolving fields in vaccine research to eradicate acquired and hereditary ailments. For a couple of years alone, several clinical and preclinical trials have been reported based on the efficacy of the mRNA vaccine platforms. While most of the early work in this field mainly focused on cancer, a number of contemporary research studies have discussed the efficacy and flexibility of mRNA to protect from lethal challenges of infectious pathogens, such as influenza virus, Zika virus, rabies virus, Ebola virus, T. gondii, and Streptococcus spp. and currently expect high demand to search mRNA-based vaccines against recently outbroken novel COVID-19 [56,68,149,150]. Though unmodified linear mRNA has potential clinical benefit, this field requires overcoming many challenges including the delivery concerns, target management, and short-time protein expression, all of which are major limitations and can be the critical drawbacks to expand this field for extensive clinical use [61]. The rapid developments in biomaterials and nanotechnology can notably assist to overcome these difficulties [61]. This review has receptively presented the advancement of mRNA structure with an improved translation and cutting-aged nanomaterials systems to increase the ability of systemic mRNA delivery. Although more improvements are required, it is promising to see that many mRNA-based vaccine approaches have already reached the clinical trial stage soon after their development [28,151,152,153]. Further experiments are still necessary to find a revolutionary model to improve mRNA stability against the cellular ribonuclease, such as mRNA circularization, and also to search for novel nanoparticles/biomaterials including gold nanoparticle with improved formulations, which can offer biocompatibility and high transfection efficiency, high selectivity and specificity, and efficient systemic in vivo delivery and long-durable gene expression. With the progression along with the proper design of nanomaterials/biomaterials, the mRNA-based vaccine approach will likely be of high demand in clinical uses for years.

## Figures and Tables

**Figure 1 vaccines-09-00244-f001:**
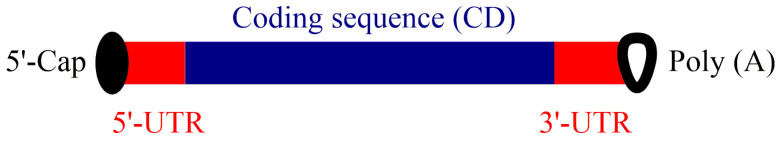
mRNA structure for optimal protein expression in vivo. An improved mRNA candidate contains 5’-cap, poly(A), 5’- and 3’-UTRs, and the coding sequence.

**Figure 2 vaccines-09-00244-f002:**
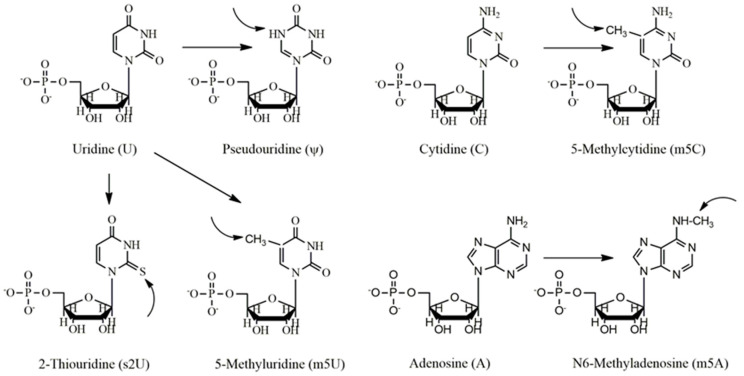
Nucleoside bases usually modified in the vaccination process. Uridine (U) is generally modified to pseudouridine (ψ), 2-thiouridine (s2U) and 5-methyluridine, cytidine (C) to 5-methylcytidine, and adenosine (A) to 5-methyladenosine (m5A).

**Figure 3 vaccines-09-00244-f003:**
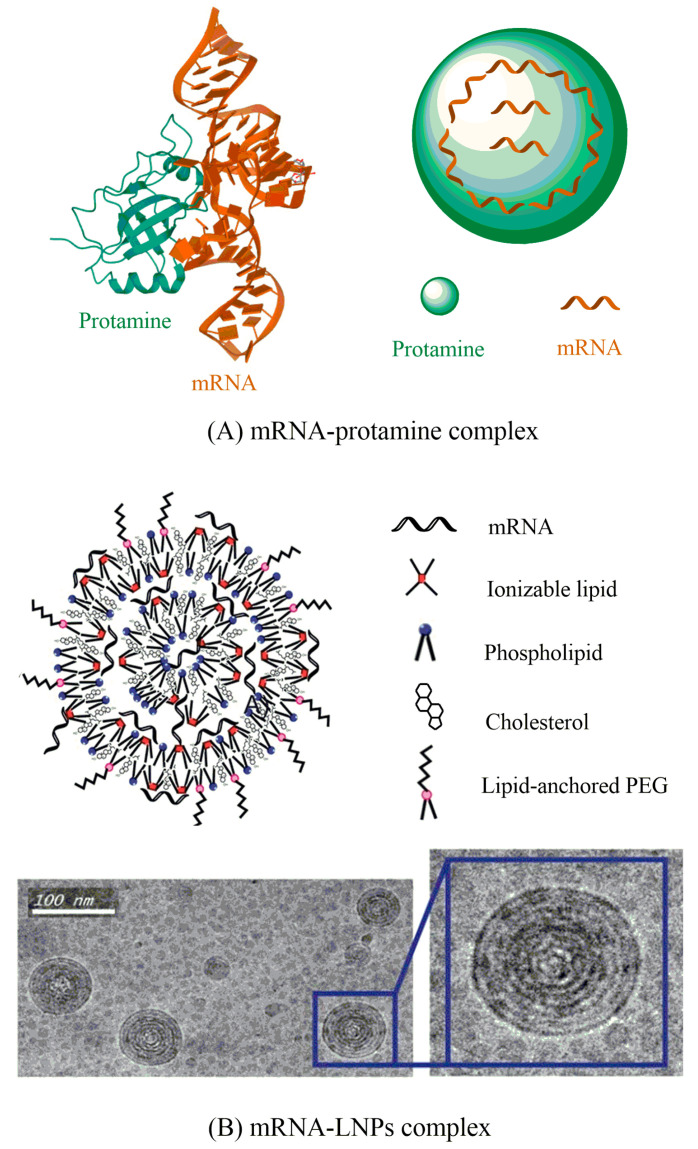
Formulations of mRNA with nanobiomaterial for in vivo drug delivery. (**A**) Protamine-complexed mRNA for drug delivery. (**B**) Synthetic components and electron microscopy images of various LNPs. LNPs have been reported to synthesize by mixing anionic mRNA with lipophilic compounds in ethanol using a microfluidic device. At lower pH, the lipid-mRNA complex can accelerate both endocytosis as well as endosomal escape. Phospholipid used during the formulation process gives structural integrity to the lipid bilayers and can contribute to the endosomal release of the mRNA to the cytoplasm. Cholesterol assists to stabilize lipid nanoparticles and stimulates membrane fusion. The lipid-coated PEG (poly-ethylene-glycol) prevents the aggregation of LNP and decreases nonspecific interactions (up). Cryogenic transmission electron microscopy image indicates that the lipid nanoparticles have a spherical shape consisting of a multilamellar structure (bottom) (Adapted with permission with little modification from [57]).

**Figure 4 vaccines-09-00244-f004:**
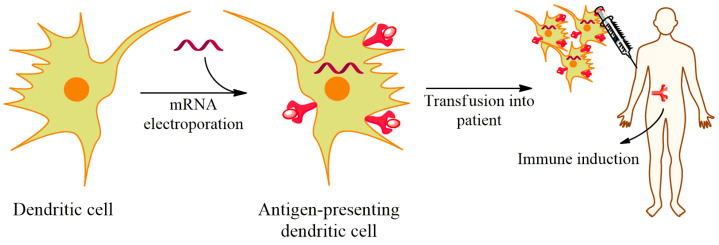
mRNA electroporation into the dendritic cell for vaccination process. mRNA transfection induces DC to present antigens, which then transfuse into the patient to establish immune defense.

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
