# Peer review of "An Overview on the Development of mRNA-Based Vaccines and Their Formulation Strategies for Improved Antigen Expression In Vivo"

_vaccines, 2021, doi:10.3390/vaccines9030244_

Round 1
Reviewer 1 Report
Rahman et al. summarized current technological improvements on mRNA-based vaccines. However, other topics should be added to this review: issues related to mRNA-based vaccines, human clinical trial on mRNA-based vaccines
Author Response
Response to Reviewer 1 Comments
Point 1: Rahman et al. summarized current technological improvements on mRNA-based vaccines. However, other topics should be added to this review: issues related to mRNA-based vaccines, human clinical trial on mRNA-based vaccines 

Response 1: Thank you for the comments. The authors have added a section as “Challenges and safety issues in the development of mRNA-based vaccines against novel antigens” regarding these topics into the revised manuscript.
Reviewer 2 Report
In this work Rahman et al., give an overview of mRNA vaccines in terms of structure of IVT mRNA and formulation of mRNA for in vivo drug delivery.
It is important right now to publish reviews of mRNA vaccines as a lot of safety issues have been raised with the new SARS-CoV-2 vaccine from Pfizer/BioNtech and Moderna.
Major point:
The manuscript needs extensive editing of English language and style.
Minor points:
The special issue of this Journal is called "The Past, Present, and Future of mRNA Vaccines" and the title of this review is "mRNA-based vaccines: the past, present and future". Maybe a different title would be appropriate, especially since this manuscript is not investigating the history of mRNA vaccines or similar. It gives an overview mRNA formulation and drug delivery.
In Chapter 2. Structure of IVT mRNA for improved translation:
- “In the cytoplasm, substituting rare codons with recurrently used identical codons that show copious cognate tRNA is a usual habit to boost mRNA translation [45], while this model has been questioned in their accuracy [46].” Please elaborate why this model has been questioned in their accuracy?
- According to the heading I don’t think the chapter “2.2 Electroporation of antigen-presenting cells (APCs)” is fitting here. The structure of the chapter should be reconsidered.
Author Response
Response to Reviewer 2 Comments
Point 1: In this work Rahman et al., give an overview of mRNA vaccines in terms of structure of IVT mRNA and formulation of mRNA for in vivo drug delivery.
Response 1: The authors appreciated your comment with thanks.
Point 2: It is important right now to publish reviews of mRNA vaccines as a lot of safety issues have been raised with the new SARS-CoV-2 vaccine from Pfizer/BioNtech and Moderna.
Response 2: Thank you for your suggestion. According to the reviewer’s instructions, the authors added a new paragraph in the last part of the manuscript as “5. Challenges and safety issues in the development of mRNA-based vaccines against novel antigens” and marked it with yellow color. While adding this section with references, some references have been changed its orders that have been marked in the MS file.
Point 3: The manuscript needs extensive editing of English language and style.
Response 3: The English of the manuscript has been extensively edited in the text and marked with yellow color.
Point 4: The special issue of this Journal is called "The Past, Present, and Future of mRNA Vaccines" and the title of this review is "mRNA-based vaccines: the past, present and future". Maybe a different title would be appropriate, especially since this manuscript is not investigating the history of mRNA vaccines or similar. It gives an overview mRNA formulation and drug delivery.
Response 4: Thank you for your suggestion regarding the title of the manuscript. Firstly, the article has been withdrawn from the special issue “The Past, Present, and Future of mRNA Vaccines” and has been processed as a regular paper in “Cellular/Molecular Immunology”. Secondly, the title has also been reconsidered with a new one “An overview on the development of mRNA-based therapeutics and their formulation strategies for improved antigen expression in vivo”.
Point 5: In Chapter 2. Structure of IVT mRNA for improved translation:
“In the cytoplasm, substituting rare codons with recurrently used identical codons that show copious cognate tRNA is a usual habit to boost mRNA translation [45], while this model has been questioned in their accuracy [46].” Please elaborate why this model has been questioned in their accuracy?
Response 5: Why substituting rare codon has been questioned in their accuracy? This answer has been elaborately explained in the text and marked in yellow color.
Point 6: According to the heading I don’t think the chapter “2.2 Electroporation of antigen-presenting cells (APCs)” is fitting here. The structure of the chapter should be reconsidered.
Response 6: Thank you for your nice comment. The authors removed the chapter and placed it as a part of formulation strategies as “4.3. Electroporation plus nanoparticles formulation”.

Round 2
Reviewer 1 Report
I appreciate that the authors have considered my recommendations for their review now suitable and complete for publication.
Author Response
Point 1: I appreciate that the authors have considered my recommendations for their review now suitable and complete for publication.
Response 1: The authors like to thank the reviewer for his valuable time to review the article
Reviewer 2 Report
I appreciate that the authors have considered my recommendations for their review. However, the manuscript still needs some editing of the English language.
Some examples which need editing:
Lane 22: Vaccines protect millions of diseases and thousands of lives each year [1]. Please change this sentence. Vaccines don't protect of diseases but from diseases.
Lane 442: Although this principle offers key benefits against several feline coronaviruses and Flavivirus, SARS-CoV and SARS-CoV-2 did not exhibit any effect in humans or non-human primates [132], which should be taken into consideration while evolving therapeutics against novel viruses [133,134]. What is the benefit of ADE against viruses? For Dengue vaccination ADE is posing a big issue.
Lane 445: Nevertheless, considering the emergency need for a COVID-19 vaccine worldwide, being troubled and evaluating such treats should not stop the process of otherwise safe and efficient vaccines to the general populations [135-137]. Sentence is difficult to read.
Lane 474: For a couple of years alone have witnessed several clinical and preclinical reports on the efficacy of the mRNA-based vaccine platforms. That sentence is missing words or grammar needs to be fixed.
Lane 451: Although mRNA vaccines can be manufactured with a minimum time, large-scale production of these therapeutics remains a challenging task with huge uncertainty to meet the demand during the pandemic. Sentence is difficult to read.
Author Response
Point 1: I appreciate that the authors have considered my recommendations for their review. However, the manuscript still needs some editing of the English language.
Response 1: The authors appreciated your comment with thanks.
Some examples which need editing:
Point 2: Lane 22: Vaccines protect millions of diseases and thousands of lives each year [1]. Please change this sentence. Vaccines don't protect of diseases but from diseases.
Response 2: The authors understood your point, modified the text and marked it with yellow color.
Point 3: Lane 442: Although this principle offers key benefits against several feline coronaviruses and Flavivirus, SARS-CoV and SARS-CoV-2 did not exhibit any effect in humans or non-human primates [132], which should be taken into consideration while evolving therapeutics against novel viruses [133,134]. What is the benefit of ADE against viruses? For Dengue vaccination ADE is posing a big issue.
Response 3: The authors mistakenly interpreted it. We revised by adding a citation. Some of the references have been changed its order that is also marked in the text.
Point 4: Lane 445: Nevertheless, considering the emergency need for a COVID-19 vaccine worldwide, being troubled and evaluating such treats should not stop the process of otherwise safe and efficient vaccines to the general populations [135-137]. Sentence is difficult to read.
Response 4: The authors slightly modified the text to make the reader understand and marked it with yellow color.
Point 5: Lane 474: For a couple of years alone have witnessed several clinical and preclinical reports on the efficacy of the mRNA-based vaccine platforms. That sentence is missing words or grammar needs to be fixed.
Response 5: The authors fixed the issue in the text and marked with yellow color.
Lane 451: Although mRNA vaccines can be manufactured with a minimum time, large-scale production of these therapeutics remains a challenging task with huge uncertainty to meet the demand during the pandemic. Sentence is difficult to read.
Response 6: The authors slightly modified the text to make the reader understand and marked it with yellow color.